# Fortification of Crude Protein Extract from Sung Yod and Hom Rajinee Rice Brans in the Development of Functional Jelly Products

**DOI:** 10.3390/foods12061138

**Published:** 2023-03-08

**Authors:** Kanthaporn Junsara, Chutha Takahashi Yupanqui, Arthitaya Kawee-ai, Rajnibhas Sukeaw Samakradhamrongthai

**Affiliations:** 1Functional Food and Nutrition Program, Faculty of Agro-Industry, Prince of Songkla University, Hat Yai 90110, Thailand; 2Center of Excellence in Functional Foods and Gastronomy, Faculty of Agro-Industry, Prince of Songkla University, Hat Yai 90110, Thailand; 3Department of Cannabis and Medicinal Plants for Local Development, Graduate School, Payap University, Chiang Mai 50000, Thailand; 4Cluster of High Value Products from Thai Rice and Plant for Health, Chiang Mai University, Chiang Mai 50100, Thailand; 5Cluster of Innovative Food and Agro-Industry, Chiang Mai University, Chiang Mai 50100, Thailand; 6Division of Product Development Technology, Faculty of Agro-Industry, Chiang Mai University, Chiang Mai 50100, Thailand

**Keywords:** crude protein extract, rice bran by-product, jelly fortification, bioactive activities, ACE-inhibitory activity, Innovative food

## Abstract

Rice bran protein (RBP) has shown good nutritional and biological values. The present study aimed to determine the functional properties of rice bran crude protein (RBCP) and apply RBCP to a rice jelly recipe to improve the jelly quality and make it an acceptable product for consumers. The design used in the jelly formulation was a central composite design. The freeze-dried crude protein of Sung Yod (SY; 0.00–0.50%) and Hom Rajinee (HR; 0.00–0.50%) rice brans were applied to the rice jelly recipe. The crude protein extract significantly influenced the physicochemical, sensory, and angiotensin I converting enzyme (ACE)-inhibitory activity of the developed jellies (*p* < 0.05). The optimized jelly contained 0.11% SY and 0.50% HR crude protein extract. The rice jelly fortified with lyophilized RBCP presented a high content of bioactive compounds (phenolic and flavonoids) with antioxidant activity and ACE-inhibitory activity. Therefore, the crude protein extract of rice brans is a potential raw material that can be used in jelly products as a cheap material to improve the jelly’s nutritional quality without affecting consumer acceptability. The outcome of the present investigation confirms that rice bran extracts may have the potential to be further exploited as ingredients in foods.

## 1. Introduction

Rice bran is a by-product of rice processing, comprising approximately 5–10% of the total weight of rice grain [1,2,3]. Rice bran is composed of various bioactive compounds including gamma-oryzanol, tocotrienols, tocopherols, phenolic acids, and flavonoids [2,3]. Rice bran also has a high content of protein (10–16%), which presents a valuable source of bioactive peptides [4]. To date, the applications of rice bran in the food industry have received increased interest from researchers due to its high nutritional value, low cost, and health benefits [2,3]. Rice bran protein (RBP) is a mixed protein and consists of albumin, globulin, glutelin, and prolamin [5]. The biological and nutritional values of RBP have been compared with protein from vegetables and animals such as soy protein, whey protein, and casein [6]. In addition, rice bran protein possesses different biological activities including antioxidant, antidiabetic, anticancer, antimicrobial, anti-inflammatory, and antihypertensive activities [6,7]. Currently, RBP has been utilized as a nutritional supplement, flavor enhancer, functional ingredient, confectionary, coffee whitener, soft drink and juice additive, cosmetic, and personal care product due to its hypoallergenic effect [8]. Ghorbani-HasanSaraei et al. [9] found that substituting fat with RBP improved the foam structure of whipped cream. The addition of RBP significantly decreased the fat content in fried fish cake, while the total phenolic content (TPC) and antioxidant activity were increased [10].

Jelly is a the sugar confectionery which consists of gelling agents, sweeteners, acids, flavoring, and color agents [11]. Despite being a product that has commonly had a high sugar content, jelly is popular and increases in manufacturing every year. In recent years, the candy market has launched a product with claims of functionality, fortified formulation, healthiness, and sustainability [12]. This was due to the increase in consumer demand for goods with convenient, nutritious, and natural foods [13]. In this study, crude protein extract from two cultivars of rice brans, namely, Sung Yod (SY) and Hom Rajinee (HR), was used in the fortification of jelly products to increase their functional properties. The two cultivars were well-known rice strains in the Phatthalung Province of Thailand. The SY strain is a red pigmented rice, while HR is a hybrid white rice from six rice varieties: Jasmine white rice, Jasmine brown rice, glutinous brown rice, Rose rice, Surin red brown rice, and black glutinous rice. Jan-On et al. [14] reported that SY rice bran hydrolysates could decrease the oxidative stress, angiotensin II, and ACE activities in Sprague-Dawley rats with no harmful effects. It could prevent hypertension and has the potential to be developed as a dietary supplement [14]. Thus, this study investigated the effect of the crude protein extracts from SY and HR on the physical, chemical, and textural properties and the sensory analysis of jelly, while a response surface methodology and central composite design (CCD) were applied to optimize the jelly’s formulation. The ACE-inhibitory activity of the fortified jelly was also evaluated.

## 2. Materials and Methods

### 2.1. Rice Brans and Crude Protein Extract

Sung Yod (SY) and Hom Rajinee (HR) rice brans were purchased from local farms in Phatthalung, Thailand in 2020. The rice brans were defatted following the method of Wang et al. [15] with slight modifications. Firstly, the rice brans (100 g) were defatted with 3-volume of 95% ethanol (300 mL) for 4 h and centrifuged at 3000 rpm for 10 min. The defatted rice brans were dried under a hood (Toplab, Bangkok, Thailand), then ground on a multi-functional high-speed grinder (Thaigrinder, Nonthaburi, Thailand) and sieved (150 µm). Crude protein extractions were modified using the method of Singh et al. [16]. The powder of the defatted rice brans was mixed with distilled water in a ratio of 1:3 (*w*/*v*) and pH-adjusted to 10.0 with 2 M NaOH. The extraction was performed at room temperature for 30 min using magnetic stirring. The collected supernatant was centrifuged at 10,000 rpm for 15 min (Hettich, Tuttlingen, Germany). The crude protein was precipitated by adjusting the pH to 4.5 using 2 M HCl and then centrifuged at 10,000 rpm for 15 min. The supernatant was then evaporated at 50 ± 2 °C until it was dry using a rotary evaporator under a vacuum. The extractant was lyophilized using a freeze dryer (Labconco Corp., Kansas City, MO, USA)) following the method of Zhao et al. [17]. The extraction yields of the HR and SY crude protein extracts were 5.1% and 5.4%, respectively. The protein content of the HR and SY rice brans was 50.89–54.63%. This result was consistent with the protein content (52.5–58.9%) of rice bran extracted by alkaline methods in a study by Chandi and Sogi [18]. The total phenolic content (TPC), total flavonoid content (TFC), and antioxidant and ACE-inhibitory activities of the lyophilized crude proteins were evaluated as presented in Table 1.

### 2.2. Experimental Design of the Suitable Quantity of Crude Protein Extract

To evaluate the interaction effects between the HR and SY crude protein extracts on the fortified jelly, augmented central composite design (CCD) was performed in Design Experts version 6.0.1 software (Stat-Ease Inc., Minneapolis, MN, USA). The content of crude protein extracts from the SY and HR rice brans, as presented in Table 2, was weighed and mixed with other ingredients for 3 min. The jelly ingredients consisted of 86% water, 10.75% sugar, 0.15% konjac powder, 2.35% carrageenan, 0.25% citric acid, and the crude protein extracts from the SY and HR rice brans. Briefly, the water was heated until the temperature reached 75 ± 2 °C. Then, the sugar, konjac powder, and carrageenan were further added and mixed for 5 min. The pH of the jelly was adjusted to 4.0 with citric acid. The supplement jelly was then placed in a refrigerator to form a gel before undergoing further analysis.

### 2.3. Physicochemical and Texture Analysis

#### 2.3.1. Moisture Content and Water Activity (a_w_)

The moisture content was measured using the method of AOAC [19]. The a_w_ was monitored using an Aqua LAB 4TEV (Decagon Devices, Inc., WA, USA). All experiments were completed in triplicates.

#### 2.3.2. Determination of Color Value (L* a* b*)

The jelly color was directly measured in terms of CIELab values (L*, a*, and b*) with a Minolta Chroma (Minolta, VA, USA). The color was measured in triplicate.

#### 2.3.3. Hardness

The hardness of the fortified jelly was analyzed by a texture analyzer (TA.XT Plus, Stable Micro Systems, Godalming, UK). The measurement was performed at room temperature and the parameters of the texture analysis instrument were a pre-test speed of 2 mm/s, distance of 3 mm, test speed of 2 mm/s, post-speed of 2 mm/s, pitch of 3 mm, and 50 kg load cell. The measurements were completed with ten replications.

#### 2.3.4. Total Phenolic Content (TPC)

Before analysis, the jelly was extracted with ethanol (1:5 *w*/*v*) using a magnetic stirrer for 15 min. The extract was then filtered through Whatman filter No. 1. The TPC was determined according to the method of Ben Rejeb et al. [20], with some modifications. An aliquot (100 µL) was mixed with 0.5 mL of Folin-Ciocalteau reagent and 2 mL of 20% (*w*/*v*) Na_2_CO_3_. After standing in dark conditions for 30 min, the mixture measured the absorbance at 750 nm. The TPC was expressed as mg of gallic acid (GAE)/100 g dry weight. The estimations were carried out in triplicates.

#### 2.3.5. Total Flavonoid Content (TFC)

The TFC of the rice jelly was determined following the method of Ben Rejeb et al. [20]. The extract (1 mL) was mixed with 1 mL of 2% aluminum trichloride solution and kept in the dark for 15 min before 415 nm of absorbance was measured. The TFC of the extract was expressed as mg of catechin equivalent (CE)/100 g dry weight. The experiments were completed in triplicate.

#### 2.3.6. Antioxidant Activities

Three methods, namely, DPPH, ABTS, and FRAP assays, were used to determine the antioxidant activities of the rice jelly [20,21]. All experiments were triplicate replications.

For the DPPH method, 200 µL of the extracted sample was mixed with 3.8 mL of 0.5 mM DPPH solution and kept in the dark for 30 min. The absorbance was measured at 517 nm and the results were expressed as mg of Trolox equivalent (TE)/100 g dry weight.

For the ABTS assay, a mixture of the extractant (150 µL) and ABTS (2.85 mL) solution (7 mM ABTS in water and 2.45 mM potassium persulfate, 1:1) was kept for 15 min before measurement at 734 nm. The results were expressed as mg of Trolox equivalent (TE)/100 g dry weight.

For the FRAP analysis, 0.25 mL of extractant was reacted with 0.25 mL of FRAP reagent (300 mM acetate buffer; pH 3.6, 10 mM 2,4,6-Tris(2-pyridyl)-s-triazine (TPTZ) in 40 mM HCl and 20 mM FeCl_3_ at a ratio of 10:1:1) for 20 min. Then, 1 mL of distilled water and 0.2 mL of 1% ferric chloride were added and measured at 593 nm. The results were expressed as mg of Trolox equivalent (TE)/100 g dry weight.

#### 2.3.7. Angiotensin I Converting Enzyme (ACE) Inhibitory Activity

The inhibitory activity of ACE was analyzed by the modified method of Lim et al. [22]. Fifty microliters of crude protein extract were mixed with 50 µL of 0.1 M sodium borate buffer (pH 8.3) and ACE and incubated at 37 °C for 5 min. Continuously, Hip-His-Leu (50 µL) was added and further incubated for 30 min. After that, 250 µL of 0.5 N HCl was added to terminate the enzyme reaction and then mixed with 1.5 mL of ethyl acetate. The mixture was then centrifuged at 6000 rpm for 10 min. The collected supernatant (1 mL) was heated at 90 °C for 5 min and mixed with 3.0 mL of distilled water before being measured at 228 nm. The ACE inhibitory activity of the samples was calculated using Equation (1). The ACE inhibitory activity was expressed as a percentage of inhibition.
(1)$$ACE\;inhibition\;\left(\%\right)=\frac{Absorbance\;control-Absorbance\;sample}{Absorbance\;control}\times 100$$

#### 2.3.8. Sensory Evaluation

This study was completed under the ethical guideline number HSc-HREC-62-25-1. The 60 (42 females and 18 males) panelists were students and staff of the University in the age range of 20–40 years old. Before participating, all participants (*n* = 60) received and signed a consent form. The jelly samples were in a disposable closed-lid plastic cup and stored at 4 °C before the assessment. The 9-point hedonic scale was used for scoring the appearance, color, texture while scooping, texture while chewing, flavor, sweetness, softness, aftertaste, and overall liking of the developed jelly.

### 2.4. Statistical Analysis

All experiments were carried out according to the relevant guidelines and regulations. The data are shown as the means and standard deviations. The differences between the SY and HR crude protein extracts were analyzed by an independent sample *t*-test. The analysis of variance (ANOVA) at *p* < 0.05 was used for comparing the statistical significances (SPSS version 17.0, Chicago, IL, USA). The experimental data were analyzed using response surface regression following the second-order polynomial model (Equation (2)):(2)$$Y=\beta_{0}+\overset{}{\underset{i=1}{\sum }}\beta_{i}X_{i}+\overset{}{\underset{i=1}{\sum }}\beta_{ii}X_{i}^{2}+\overset{}{\underset{i=1}{\sum }}\overset{}{\underset{j=i+1}{\sum }}\beta_{ij}X_{i}X_{j}+e_{0},\;\;\;$$
where *Y* is the predicted response variable, $\beta_{0}$ is the constant coefficient, $\beta_{i}$ is the linear effect, $\beta_{ii}$ is the squared effect, $\beta_{ij}$ is the interaction effect, and $X_{i}$ and $X_{j}$ represent the independent variables, respectively.

## 3. Results and Discussion

### 3.1. Properties of the SY and HR Crude Protein Extracts

The TPC, TFC, antioxidant activity, and ACE-inhibitory activity of the crude protein extracts from the SY and HR rice brans are shown in Table 1. The SY crude protein extract had higher contents of TPC (5.36-fold) and TFC (5.23-fold), as well as greater antioxidant activities (1.16–2.22-fold) than those of the HR crude protein extract. This might have been due to the husk of SY being red-brown, as this color indicates the presence of polyphenolic compounds, proanthocyanin, anthocyanin, and flavonoids [14]. However, the SY crude protein extract exhibited a lower ACE-inhibitory activity than the HR crude protein extract by 1.7%. The good ACE inhibitory activity might have been due to the high proportion of positively charged amino acids such as lysine and arginine at the C-terminal position [23]. Therefore, the higher ACE-inhibitory activity of the HR crude protein extract might have indicated the presence of lysine and arginine at the C-terminal position in their structures.

These findings revealed that both crude protein extracts possess potential benefits, with different biological activities.

### 3.2. Effect of the Crude Protein Extracts on the Properties of the Fortified Jelly

#### 3.2.1. Physical Properties of the Fortified Jelly Using the Crude Protein Extracts from the Sung Yod and Hom Rajinee Rice Brans

The fortification of the SY and HR crude protein extracts significantly affected the physical properties of the jelly (Table 2). There were slight changes in the moisture content (84.87 ± 0.21–86.40 ± 0.05%) and water activity (0.960 ± 0.052–0.994 ± 0.003) of the jelly. The change in the contents of moisture and a_w_ might have been due to the decomposition of the water molecules into oxygen radicals [24]. The addition of the SY and HR crude protein extracts significantly decreased the L* value (*p* < 0.05) in comparison with the control (Formula 5), indicating that those samples were darker in comparison with the jelly without a crude protein extract addition. Meanwhile, the redness (a*) and yellowness (b*) increased with the increasing crude protein extracts. A high content of the SY crude protein extract presented a high a* and b* values. This was due to SY containing several pigment compounds, which were red-brown, and which responded to the color alternation of the jelly products. Noticeable visual changes in color were also determined when citrus fruits were added to the jelly [20].

Texture property is an important property of jelly. In this study, the gel strength of the fortified jellies was measured (Table 2). The gel strengths of the fortified jellies (0.19 ± 0.00–0.27 ± 0.01 N) were significantly different (*p* < 0.05) from each other and were lower than that of the control (0.33 ± 0.00 N). However, the addition of 0.50% HR crude protein extract increased the gel strength of the jelly to 0.51 ± 0.01 N. This may be attributed to the intrinsic molecules and amino acid degradation, which was affected by the formation of peptides. When the proteins were subjected to heat or pH adjustment, the proteins were denatured and aggregated into a network structure, which resulted in the formation of a protein gel [25].

#### 3.2.2. Chemical Properties of the Fortified Jelly Using the Crude Protein Extracts from the Sung Yod and Hom Rajinee Rice Brans

The TPC and TFC of the fortified jelly were recapitulated as shown in Table 3. The values ranged between 372 ± 18 and 4944 ± 87 mg GAE/100 g sample and 96 ± 6 and 1203 ± 42 mg CE/100 g sample, respectively, which was higher than that of jelly without crude protein extract addition (Formula 5). This could be attributed to the polyphenolic and flavonoid contents of the SY and HR crude protein extracts (Table 1). DPPH, ABTS, and FRAP are popular assays used to evaluate the antioxidant activity of natural products [21]. The antioxidant activities of the fortified jellies increased in the same trends as the TPC and TFC values, indicating a positive relationship between these values. This noticeably agreed with the addition of citrus fruits [20] and dandelion leaves polysaccharide extract [26] in the jelly.

Meanwhile, the ACE-inhibitory activity of the fortified jelly increased with the increasing contents of the SY and HR crude protein extracts (Table 3). The highest ACE-inhibitory activity (46.46 ± 0.25%) of the fortified jelly was found for the combination of SY and HR at 0.50 plus 0.50%. This value was higher than those of the fortified jellies with the SY and HR crude protein extracts alone by 48.71% for HR and 52.32% for SY. Based on these results, fortified jelly may be a health benefit and prevent oxidative stress-related diseases [26].

#### 3.2.3. Sensory Evaluation of the Fortified Jelly Using the Crude Protein Extracts from the Sung Yod and Hom Rajinee Rice Brans

The sensory scores revealed that appearance, color, texture while scooping, texture while chewing, aftertaste, and overall liking were significantly different at *p* < 0.05 (Table 4). However, insignificant differences in flavor, sweetness, and softness were detected. The appearance scores of the fortified jellies (6.0 ± 1.5–6.9 ± 1.1) were significantly lower than the control (7.1 ± 1.1), indicating that the panelists preferred the appearance of the control jelly to the fortified jellies. This might have been due to the changes in color, which affected the visual appearance of the fortified jellies. The higher scores for textures while scooping (7.3 ± 0.9) and in the mouth (7.3 ± 1.0) of the 0.50% HR jelly were due to the form of protein gel, which resulted in increased gel strength, which was preferred by the panelists. A lower score for the aftertaste of the fortified jelly was due to the bitter taste of the crude protein extract, which contained bitter phenolic compounds [26]. Average scores for overall liking of the fortified jellies ranged from 6.1 ± 1.0 to 7.0 ± 1.1 on a nine-point hedonic scale. The addition of 0.5% HR crude protein extract seemed to satisfy the panelists’ preferences by scoring for overall liking to 7.0 ± 1.1, which is higher than the control (6.6 ± 1.4).

### 3.3. Model Fitting for the Fortified Jelly Using the Crude Protein Extracts from the Sung Yod and Hom Rajinee Rice Brans

All investigated parameters (21 variables) were utilized to predict the linear and quadratic interactions of the SY and HY crude protein extracts using response surface models. It was found that sixteen parameters were the significant variables (Table 5 and Figure 1).

The amount of SY crude protein extract showed a negative sign on the L* value and all sensory attributes, which means that the addition of this substance could decrease the lightness of the jelly and affect consumer perceptions. The increase in SY crude protein extract significantly decreased the L* value (Figure 1a), appearance (Figure 1j), color (Figure 1k), texture in mouth (Figure 1l), sweetness (Figure 1m), sourness (Figure 1n), aftertaste (Figure 1o), and overall liking (Figure 1p), while the a* and b* values and the chemical properties (TPC, TFC, DPPH, ABTS, FRAP, and ACE-inhibitory activity) presented positive effects of the SY crude protein extract. Thus, the increase in SY crude protein extract concentration in the jelly significantly increased the values of a* (Figure 1b), b* (Figure 1c), TPC (Figure 1d), TFC (Figure 1e), DPPH (Figure 1f), ABTS (Figure 1g), FRAP (Figure 1h), and ACE-inhibitory activity (Figure 1i). On the other hand, the addition of the HR crude protein extract showed negative effects on L*, a*, appearance, color, sweetness, and sourness. However, the HR crude protein extract presented positive signs to b*, chemical properties, texture in mouth, aftertaste, and overall liking. Therefore, the increase in HR crude protein extract also exhibited a positive correlation to the chemical properties and triggered the consumer preference for texture in mouth, aftertaste, and overall liking. Thus, in conclusion, the addition of the SY and HR crude protein extracts in jelly could increase the jelly’s bioactive compounds, which provides benefits for human health.

### 3.4. Model Validation of the Fortified Jelly Using the Crude Protein Extracts from the Sung Yod and Hom Rajinee Rice Brans

The addition of the SY crude protein extract positively increased the TPC, TFC, DPPH, ABTS, FRAP, and ACE-inhibitory activity of the fortified jelly. Moreover, the results of the sensory evaluation showed that the consumers preferred the jelly with 0.5% HR crude protein extract (7.0 ± 1.1) over the other formulations. Thus, the optimum concentrations of SY and HR crude protein extracts were required to reach the highest values of physicochemical (L*, a*, b*, TPC, TFC, DPPH, ABTS, FRAP, and ACE-inhibitory activity) and sensory (appearance, color, texture in mouth, sweetness, sourness, aftertaste, and overall liking) scores, and those for other variables such moisture content, a_w_, gel strength, texture while scooping, and flavor attributes (in a range) were determined by solving the equation in Table 5 using Design-Expert software. The SY and HR values were estimated at 0.11% and 0.50%, respectively. The predicted and experimental values of the optimal fortified jelly are presented in Table 6.

Under the optimal concentrations, the fortified jelly showed L*, a*, b*, TPC, TFC, DPPH, ABTS, FRAP, and ACE-inhibitory activity, as well as appearance, color, texture in mouth, sweetness, sourness, aftertaste, and overall liking scores of 39.75 ± 0.07, 3.61 ± 0.11, 17.32 ± 0.20, 2398.75 ± 55.32 mg GAE/100 g DW, 677.79 ± 23.32 mg CE/100 g DW, 169.97 ± 11.21 mg TE/100 g DW, 895.45 ± 26.22 mg TE/100 g DW, 833.98 ± 11.78 mg TE/100 g DW, 28.85 ± 0.55%, 6.93 ± 1.0, 6.85 ± 0.9, 6.51 ± 1.3, 6.64 ± 1.2, 6.78 ± 0.9, 6.83 ± 1.0, and 6.87 ± 0.8, respectively. The approximation errors between the predicted and experimental values of the significant variables of the fortified jelly were in the range of 0.29–39.34%.

The sensorial scores of the fortified jelly were approximately 6 to 7, which was in the range of ‘like slightly’ to ‘like moderately’. Furthermore, the overall liking scores of the fortified jelly were 6.87 ± 0.8, which was higher than that of the control (6.6 ± 1.4, Table 4). Thus, the addition of the SY and HR crude protein extracts at 0.11% and 0.50% was acceptable for jelly formulation.

## 4. Conclusions

The findings from this research concluded that agricultural by-products akin to rice bran in terms of using crude protein extracts as an ingredient in jelly formulations indicated positive influences on the jelly’s chemical and sensory qualities. Specifically, the SY and HR crude protein extracts remarkably enhanced the TPC, TFC, antioxidant activities, and ACE-inhibitory activities of the jelly product. In particular, the panelists were more satisfied with HR crude protein extract. Furthermore, based on several sensory attributes, the jelly combined with the crude protein extracts was acceptable to the test panelists. Therefore, the bioactive compounds of rice bran can be considered as an added-value component in the production of jelly that provides functional enrichment.

## Figures and Tables

**Figure 1 foods-12-01138-f001:**
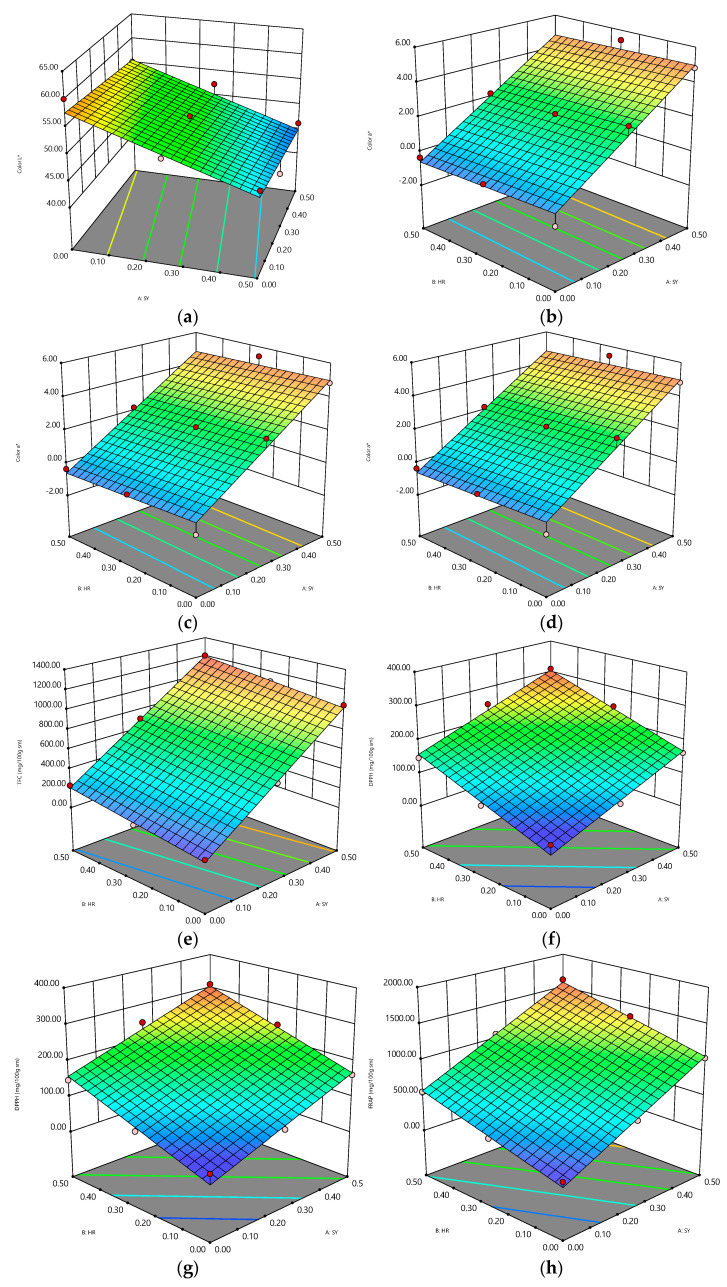
The contour plots of the interaction effects of the Sung yod (A) and Hom Rajinee (B) crude protein extracts on the jelly properties: L* (**a**), a* (**b**), b* (**c**), TPC (**d**), TFC (**e**), DPPH (**f**), ABTS (**g**), FRAP (**h**), ACE-inhibition (**i**), appearance (**j**), color (**k**), texture in mouth (**l**), sweetness (**m**), sourness (**n**), aftertaste (**o**), and overall liking (**p**).

**Table 1 foods-12-01138-t001:** Total phenolic, total flavonoids, antioxidant activity, and anti-angiotensin inhibitory activity of the crude protein extracts from the Sung Yod and Hom Rajinee rice brans ^1^.

Parameters	SY Crude Protein Extract	HR Crude Protein Extract
TPC (mg GAE/g)	81.78 ± 1.48 ^a^	15.62 ± 0.63 ^b^
TFC (mg CE/g)	20.66 ± 0.02 ^a^	3.95 ± 0.08 ^b^
Antioxidant activities		
—ABTS (mg TE/g)	22.13 ± 0.53 ^a^	9.98 ± 0.33 ^b^
—DPPH (mg TE/g)	3.28 ± 0.03 ^a^	2.81 ± 0.01 ^b^
—FRAP (mg TE/g)	21.02 ± 0.24 ^a^	10.72 ± 0.15 ^b^
ACE inhibitory activity (%)	22.26 ± 0.47 ^b^	23.96 ± 0.64 ^a^

^1 a–b^ within the same row denotes a statistical difference at *p* < 0.05 determined by *t*-tests.

**Table 2 foods-12-01138-t002:** Effects of the crude protein extracts from the Sung Yod and Hom Rajinee rice brans on the moisture content, water activity (a_w_), color, and gel strength ^1^ of the fortified jelly.

Formula	Factors (%)		Moisture (%)	a_w_ ^ns 2^	Color			Gel Strength (N)
	SY (A)	HR (B)			L*	a*	b*	
1	0.00	0.50	84.95 ± 0.23 ^e^	0.990 ± 0.001	55.21 ± 0.15 ^b^	−0.35 ± 0.07 ^f^	19.55 ± 0.15 ^e^	0.51 ± 0.01 ^a^
2	0.25	0.25	85.45 ± 0.16 ^cd^	0.989 ± 0.002	52.37 ± 0.42 ^d^	2.21 ± 0.32 ^e^	22.40 ± 0.28 ^b^	0.24 ± 0.01 ^de^
3	0.00	0.25	85.50 ± 0.20 ^cd^	0.988 ± 0.001	54.52 ± 0.14 ^c^	−0.39 ± 0.10 ^f^	20.41 ± 0.26 ^d^	0.23 ± 0.00 ^de^
4	0.50	0.50	85.74 ± 0.22 ^bc^	0.989 ± 0.001	46.18 ± 0.12 ^h^	4.07 ± 0.03 ^c^	23.67 ± 0.04 ^a^	0.26 ± 0.00 ^c^
5	0.00	0.00	85.01 ± 0.13 ^e^	0.988 ± 0.001	60.14 ± 0.11 ^a^	−1.08 ± 0.08 ^g^	18.57 ± 0.15 ^f^	0.33 ± 0.00 ^b^
6	0.25	0.25	85.36 ± 0.16 ^d^	0.989 ± 0.002	49.26 ± 0.23 ^f^	2.19 ± 0.18 ^e^	21.41 ± 0.08 ^c^	0.24 ± 0.01 ^d^
7	0.25	0.50	86.01 ± 0.24 ^b^	0.960 ± 0.052	52.37 ± 0.42 ^d^	2.21 ± 0.32 ^e^	22.40 ± 0.28 ^b^	0.20 ± 0.00 ^f^
8	0.25	0.00	85.68 ± 0.15 ^bcd^	0.994 ± 0.003	51.56 ± 0.28 ^e^	2.83 ± 0.31 ^d^	22.31 ± 0.18 ^b^	0.22 ± 0.01 ^e^
9	0.50	0.25	84.87 ± 0.21 ^e^	0.989 ± 0.001	43.26 ± 0.32 ^i^	5.41 ± 0.43 ^a^	22.52 ± 1.29 ^b^	0.27 ± 0.01 ^c^
10	0.50	0.00	86.40 ± 0.05 ^a^	0.992 ± 0.003	48.06 ± 0.03 ^g^	4.83 ± 0.07 ^b^	23.53 ± 0.12 ^a^	0.19 ± 0.00 ^f^

^1 a–i^ within the same column denotes a statistical difference at *p* < 0.05 by Duncan multiple range tests. ^2 ns^ denotes non-significance (*p* > 0.05).

**Table 3 foods-12-01138-t003:** Effects of the crude protein extracts from the Sung Yod and Hom Rajinee rice brans on TPC, TFC, and antioxidant and ACE-inhibitory activities ^1^ of the fortified jelly.

Formula	Factors (%)		TPC (mg GAE/100 g Sample)	TFC (mg CE/100 g Sample)	Antioxidant Activities			ACE-Inhibitory Activity/100 g Jelly (%)
	SY (A)	HR (B)			DPPH (mg TE/100 g Sample)	ABTS (mg TE/100 g Sample)	FRAP (mg TE/100 g Sample)	
1	0.00	0.50	719 ± 97 ^g^	229 ± 23 ^g^	147 ± 4 ^c^	537 ± 23 ^f^	553 ± 2 ^f^	23.83 ± 0.64 ^c^
2	0.25	0.25	2414 ± 44 ^e^	599 ± 20 ^e^	153 ± 15 ^c^	805 ± 22 ^e^	801 ± 19 ^e^	23.02 ± 0.11 ^d^
3	0.00	0.25	372 ± 18 ^h^	96 ± 6 ^h^	78 ± 3 ^d^	243 ± 15 ^g^	278 ± 9 ^h^	11.93 ± 0.32 ^f^
4	0.50	0.50	4944 ± 87 ^a^	1203 ± 42 ^a^	310 ± 5 ^a^	1586 ± 26 ^a^	1616 ± 18 ^a^	46.46 ± 0.25 ^a^
5	0.00	0.00	332 ± 20 ^h^	55 ± 11 ^h^	48 ± 6 ^e^	532 ± 18 ^f^	116 ± 7 ^e^	0.00 ± 0.00 ^h^
6	0.25	0.25	2472 ± 41 ^e^	606 ± 8 ^e^	156 ± 12 ^c^	818 ± 12 ^e^	812 ± 5 ^i^	23.19 ± 0.11 ^d^
7	0.25	0.50	2829 ± 72 ^d^	707 ± 9 ^d^	250 ± 16 ^b^	1044 ± 11 ^d^	1071 ± 9 ^c^	34.82 ± 0.42 ^b^
8	0.25	0.00	2103 ± 52 ^f^	511 ± 2 ^f^	83 ± 5 ^d^	555 ± 10 ^f^	519 ± 7 ^g^	11.17 ± 0.24 ^g^
9	0.50	0.25	4328 ± 52 ^b^	1101 ± 31 ^b^	244 ± 1 ^b^	1346 ± 21 ^b^	1330 ± 12 ^b^	34.32 ± 0.18 ^b^
10	0.50	0.00	4192 ± 82 ^c^	1048 ± 43 ^c^	162 ± 6 ^c^	1107 ± 5 ^c^	1032 ± 17 ^d^	22.15 ± 0.47 ^e^

^1 a–i^ within the same column denotes a statistical difference at *p* < 0.05 by Duncan multiple range tests.

**Table 4 foods-12-01138-t004:** Effects of the crude protein extracts from the Sung Yod and Hom Rajinee rice brans on the sensory qualities of the fortified jelly ^1^.

Formula	Factors (%)		Appearance	Color	Texture While Scooping	Texture While Chewing	Flavor ^ns 2^	Sweetness ^ns^	Sourness ^ns^	Aftertaste	Overall Liking
	SY (A)	HR (B)									
1	0.00	0.50	6.9 ± 1.1 ^ab^	7.2 ± 0.8 ^a^	7.3 ± 0.9 ^a^	7.3 ± 1.0 ^a^	6.9 ± 1.0	6.8 ± 1.0	6.9 ± 1.1	7.0 ± 0.9 ^a^	7.0 ± 1.1 ^a^
2	0.25	0.25	6.6 ± 1.1 ^bcd^	6.8 ± 0.9 ^abc^	6.8 ± 0.8 ^b^	6.7 ± 1.1 ^bc^	6.5 ± 1.0	6.4 ± 1.1	6.4 ± 1.2	6.5 ± 1.1 ^b^	6.4 ± 1.3 ^bc^
3	0.00	0.25	6.7 ± 1.1 ^ab^	7.0 ± 1.2 ^ab^	6.7 ± 1.1 ^bc^	6.6 ± 1.0 ^bc^	6.5 ± 0.9	6.4 ± 0.9	6.5 ± 1.0	6.4 ± 0.9 ^b^	6.5 ± 1.0 ^bc^
4	0.50	0.50	6.1 ± 1.5 ^cd^	6.5 ± 1.2 ^c^	6.4 ± 1.0 ^bc^	6.4 ± 0.9 ^bc^	6.7 ± 1.1	6.5 ± 0.9	6.6 ± 1.0	6.4 ± 0.9 ^b^	6.4 ± 0.9 ^bc^
5	0.00	0.00	7.1 ± 1.1 ^a^	7.2 ± 1.0 ^a^	6.8 ± 1.2 ^b^	6.8 ± 1.2 ^b^	6.6 ± 1.2	6.4 ± 1.0	6.5 ± 1.0	6.5 ± 1.2 ^b^	6.6 ± 1.4 ^b^
6	0.25	0.25	6.3 ± 1.6 ^bcd^	6.7 ± 1.1 ^bc^	6.2 ± 1.3 ^c^	6.3 ± 1.2 ^bc^	6.2 ± 1.1	6.2 ± 1.0	6.2 ± 1.4	6.2 ± 1.2 ^b^	6.3 ± 1.1 ^bc^
7	0.25	0.50	6.3 ± 1.4 ^bcd^	6.8 ± 1.0 ^abc^	6.5 ± 0.9 ^bc^	6.5 ± 1.0 ^bc^	6.6 ± 0.8	6.5 ± 0.7	6.6 ± 0.7	6.5 ± 0.9 ^b^	6.7 ± 0.8 ^ab^
8	0.25	0.00	6.7 ± 1.2 ^abc^	6.9 ± 0.8 ^abc^	6.7 ± 0.9 ^bc^	6.6 ± 1.0 ^bc^	6.5 ± 0.8	6.3 ± 0.8	6.6 ± 0.9	6.4 ± 0.9 ^b^	6.4 ± 0.9 ^bc^
9	0.50	0.25	6.2 ± 1.5 ^bcd^	6.6 ± 1.2 ^c^	6.4 ± 1.3 ^bc^	6.4 ± 1.3 ^bc^	6.4 ± 1.0	6.4 ± 1.0	6.4 ± 1.0	6.2 ± 1.1 ^b^	6.3 ± 1.2 ^bc^
10	0.50	0.00	6.0 ± 1.5 ^d^	6.5 ± 1.1 ^c^	6.4 ± 0.9 ^bc^	6.3 ± 1.2 ^c^	6.5 ± 0.9	6.5 ± 0.9	6.4 ± 0.9	6.2 ± 0.9 ^b^	6.1 ± 1.0 ^c^

^1 a–d^ within the same column denotes a statistical difference at *p* < 0.05 by Duncan multiple range tests. ^2 ns^ denotes non-significance (*p* > 0.05).

**Table 5 foods-12-01138-t005:** Regression models of the significant variables of the fortified jelly using the crude protein extracts from the Sung Yod and Hom Rajinee rice brans ^1^.

Variables	Model	R^2^	*p*-Value
Physical properties			
L*	57.68 − 21.58A − 3.99B	0.8535	<0.01
a*	−0.39 + 10.75A − 0.43B	0.9635	<0.01
b*	19.61 + 7.46A + 0.81B	0.8307	<0.01
Chemical properties			
TPC (mg GAE/100 g)	152.98 + 8027.56A + 1243.33B	0.9960	<0.01
TFC (mg CE/100 g)	32.94 + 1980.89A + 350.00B	0.9980	<0.01
DPPH (mg TE/100 g)	20.07 + 295.78A + 276.22B	0.9723	<0.01
ABTS (mg TE/100 g)	240.93 + 1817.56A + 647.78B	0.9022	<0.01
FRAP (mg TE/100 g)	45.46 + 2020.89A + 1048.89B	0.9939	<0.01
ACE inhibition (%)	−0.07 + 44.78A + 47.86B	0.9999	<0.01
Sensory properties			
Appearance	6.98 − 1.62A − 0.29B	0.8590	<0.01
Color	7.15 − 1.20A − 0.09B	0.9262	<0.01
Texture in mouth	6.76 − 1.03A + 0.36B	0.5998	0.04
Sweetness	6.40 − 0.87A − 0.14B + 2.09A^2^ + 1.83B^2^ −1.60AB	0.7845	0.04
Sourness	6.58 − 0.33A − 1.31B + 3.38B^2^	0.5352	0.10
Aftertaste	6.49 − 0.73A + 0.53B	0.6773	0.02
Overall liking	6.54 − 0.91A + 0.64B	0.8176	<0.01

^1^ A denotes the Sung Yod crude protein extract and B denotes the Hom Rajinee crude protein extract.

**Table 6 foods-12-01138-t006:** Predicted and experimental values of the optimized jelly fortified with the crude protein extracts from the Sung Yod and Hom Rajinee rice brans.

Characteristic	Predicted Value	Experimental Value	% Error
L*	51.29	39.75 ± 0.07	22.50
a*	2.19	3.61 ± 0.11	39.34
b*	21.68	17.32 ± 0.20	20.11
TPC (mg GAE/100 g DW sample)	2470.70	2398.75 ± 55.32	2.91
TFC (mg CE/100 g DW sample)	615.67	677.79 ± 23.32	9.17
DPPH (mg TE/100 g DW sample)	163.07	169.97 ± 11.21	4.06
ABTS (mg TE/100 g DW sample)	857.27	895.45 ± 26.22	4.26
FRAP (mg TE/100 g DW sample)	812.90	833.98 ± 11.78	2.53
ACE inhibition (%)	23.09	28.85 ± 0.55	19.97
Appearance	6.50	6.93 ± 1.0	6.20
Color	6.83	6.85 ± 0.9	0.29
Texture in mouth	6.59	6.51 ± 1.3	1.21
Sweetness	6.29	6.64 ± 1.2	5.27
Sourness	6.38	6.78 ± 0.9	5.90
Aftertaste	6.44	6.83 ± 1.0	5.71
Overall liking	6.48	6.87 ± 0.8	5.68

## Data Availability

The data generated during the current study are available from the corresponding author upon reasonable request.

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
