# Peer review of "Fortification of Crude Protein Extract from Sung Yod and Hom Rajinee Rice Brans in the Development of Functional Jelly Products"

_foods, 2023, doi:10.3390/foods12061138_

Round 1

Reviewer 1 Report

This article investigated the effect of addition of two rice bran protein extractions in jelly. The results show that adding suitable rice bran proteins from SY and HR can increase some functional activities and sensory qualities. It provides some useful reference for developing the by-product of rice bran. Furthermore, the design and presentation of this article are clear. 

Of course, there are some errors or what need to be improved.

1. In abstract, Sung Yod. In title, Song Yod. Maybe writing error.

2.  3.2.2. Chemical properties of  ?

3. Line 55,  The author said jelly is unhealthy products. This point is controversial, and you can't say this simply.

Reviewer 2 Report

This article investigated the effect of rice bran crude protein extract from two known cultivars in Thailand; namely Sang Yod (SY) and Home Rajini (HE) on the physical, chemical, textural, antioxidant and antihypertensive properties of a fortified jelly recipe to help improve its sensory qualities and increase its functional properties. In addition, while using a quadratic model, central composite design (CCD) was applied to help optimize the jelly formulation.

 As a general opinion, and even if I'm not a Native English speaker, English here is a real drawback and a major flaw for this manuscript. 

More particularly

- in mat. and Meth.

-alkaline extraction is the method of choice for protein extraction from rice bran, authors must report yields of extractions,

- more over may you discuss if your extraction method is economically viable? Compare your data to the publication of Chandi and Sogi,. Functional Properties of Rice Bran Protein Concentrates. J. Food Eng. 2007, 79, 592–597 or with most recent “cho et al, in molecules (2022): Functional Properties of Rice Bran Proteins Extracted from Low-Heat-Treated Defatted Rice Bran” which you must include into the discussion.

- the authors talk also about a protein extraction method inspired by a reference from wang. 2014, which they modify slightly, however, apart from delipidation, the methods recommended by wang and authors are not the same,

- how can you justify that for both TPC and TFC analysis, you have the same weighting (5.23 times) for the SY crude protein extract compared to the HR crude protein extract?

- correct references 3, 6, 13, 15, 18

Among many more grammar and conjugation errors, typos:

- the title needs to be rephrased

- in the abstract correct barn by bran

- correct ‘’The crude protein extract influenced the physical-chemical, sensory, and angiotensin I converting enzyme (ACE)-inhibitory activity of the developed jellies (p<0.05).’’

in the introduction

- correct the paragraph : « Rice bran is a by-product from rice processing, which included 5-10% of total weight of rice gain [1-3].

- Rice bran composed of various bioactive compounds including gamma- oryzanol, tocotrienols, tocopherols, phenolic acids, and flavonoids [2,3]. »

correct:

- In this study Rice brans crude protein extract from two cultivars; namely Sang Yod (SY) and Hom Rajinee (HR) was used to fortify in jelly in order to its increase functional properties

- correct :

« rice bran protein processes different biological activities including antioxidant, antidiabetic, anticancer, antimicrobial, anti-inflammatory, and antihypertensive [6,7] »

- correct

« Ghorbani‐HasanSaraei, et al. [9] substituted fat by RBP can improved the foam structure of whipped cream »

- « Jan-On, et al. [14] reported that SY rice bran hydrolysates could be decreased oxidative stress, angiotensin II, and ACE activity of Sprague-Dawley rats with no harmful. »

- ‘’It could be prevent hypertension’’

In material and methods

- line 107 : For cielab color measurements “The color was measured triplication”

correct :

- line 144. The inhibition of ACE was modified the method of Lim, et al. [20].

- P5, line 190: Physical properties of fortified jelly

- line 216: 3.2.2. Chemical properties of ?

- lines 238/239: Rephrase “The sensory scores revealed that appearance, color, texture while scooping, texture while chewing, aftertaste, and overall liking was significant difference at p < 0.05

- Rephrase the paragraph : « The sensory scores revealed that appearance, color, texture while scooping, texture while chewing, aftertaste, and overall liking was significant difference at p < 0.05 (Table 4) »

- line 243 : color which affected to the visual appearance of the fortified jelly.

- line 289 : rephrase, ‘’Thus, in conclusion, the addition of SY and HR crude protein extracts in jelly could be increased the bioactive compounds, which had a potential for human health and benefits.’’

- lines 294-297. Rephrase and correct English : ‘’’Even though the consumers preferred jelly with 0.5% HR crude protein extract (7.0±1.1) over that other combination formulations. However, the fact, the addition of SY crude protein extract could be positively increased TPC, TFC, DPPH, ABTS, FRAP, and ACE-inhibitory activity of the jelly. Thus, the optimum concentrations of SY and HR crude protein extract to reach the highest L*, a*, b*, TPC, TFC, DPPH, ABTS, FRAP, and ACE- inhibitory activity, appearance,

……

Reviewer 3 Report

In this manuscript, the authors researched physicochemical, sensory, and functional properties of the rice jelly added with rice bran protein. Finding of authors are thought to be of some use to the rice bran. However, I think it will be a better paper if following are revised.

Revision points;

1. Abstract

- line 18: barn -> bran

- It is necessary to explain what kind of rice bran is best to add to the conclusion.

2. Introduction

- Need to explain the functionality evaluation of jelly in the last part of the research goal.

3. Materials and Methods

- line 73: Need to explain the harvest year of Sang Yod and Hom Rajine

- line 76: Need to explain the concentration of ethanol

- lin3 155: Need to explain the characteristics of panelists (age, gender, etc.)

4. Results and discussion

- line 181: Need to discuss the positively charged amino acid content of SY and HR. Need to explain why the ACE-inhibitory activity of HR was significantly high.

- line 204: What is the difference between formula 2 and 6? The formula is the same, but the results of Tables 2 and 3 are significantly different. The order of formula needs to be rearranged around SY content. Complex arrangement makes it difficult to understand data.

- line 216: Need to review the sub-headings, <chemical properties of> ?

- line 232: Need to change the title of Table 3. It's a characteristic of Jelly, but it's similar to the title of Table 1 because there's no content.

- line 248: The term "overall acceptability" and the term "overall liking" used in Table 4 are different, causing confusion, so the terminology needs to be unified.

Round 2

Reviewer 2 Report

line 136: correct: All experiments were triplicated replications.

Line 204: There was a slightly changed in the content of moisture

rephrase in the conclusion line 335 : "The results of this research showed the potential of the application of agricultural byproduct (rice bran) as a crude protein extract on the jelly formulation".

Author Response

Response to reviewer 2

A Revised Manuscript

Journal: Foods

Manuscript ID: foods-2173420

Manuscript Title: Fortification of Crude Protein Extract from Sung Yod and Hom Rajinee Rice Brans in the Development of Functional Jelly Product

Corresponding author: Rajnibhas Sukeaw Samakradhamrongthai

Email: rajnibhas.s@cmu.ac.th

line 136: correct: All experiments were triplicated replications.

Response to reviewer: Thank you very much for your comment and suggestion. The revision has been made as suggested.

“There was a slightly changed in moisture content” (Line 138)  

Line 204: There was a slightly changed in the content of moisture

Response to reviewer: Thank you very much for your comment and suggestion. The revision has been made as suggested.

“There was a slightly changed in moisture content”  (Line 206) 

rephrase in the conclusion line 335: "The results of this research showed the potential of the application of agricultural byproduct (rice bran) as a crude protein extract on the jelly formulation".

Response to reviewer: Thank you very much for your comment and suggestion. The revision has been made as suggested.

“The finding from this research can be concluded that agricultural by-products akin to rice bran in terms of using crude protein extract to be an ingredient of the jelly formulations which indicated a positive influence on chemical and sensory qualities.”

(Line 337–339) 
